# Prenatal Manganese Exposure and Long-Term Neuropsychological Development at 4 Years of Age in a Population-Based Birth Cohort

**DOI:** 10.3390/ijerph17051665

**Published:** 2020-03-04

**Authors:** Ainara Andiarena, Amaia Irizar, Amaia Molinuevo, Nerea Urbieta, Izaro Babarro, Mikel Subiza-Pérez, Loreto Santa-Marina, Jesús Ibarluzea, Aitana Lertxundi

**Affiliations:** 1Faculty of Psychology of the University of the Basque Country, 20018 San Sebastian, Spain; 2Biodonostia Health Research Institute, Group of Environmental Epidemiology and Child Development, Doctor Begiristain, s/n, 20014 San Sebastian, Spain; a-irizarloibide@euskadi.eus (A.I.); au-molinuevo@euskadi.eus (A.M.); n-urbietamacazaga@euskadi.eus (N.U.); i-babarro@euskadi.eus (I.B.); m-subiza@euskadi.eus (M.S.-P.); ambien4ss-san@euskadi.eus (L.S.-M.); mambien3-san@euskadi.eus (J.I.); aitana.lertxundi@ehu.eus (A.L.); 3Spanish Consortium for Research on Epidemiology and Public Health (CIBERESP), Instituto de Salud Carlos III, C/Monforte de Lemos 3-5, 28029 Madrid, Spain; 4Health Department of the Basque Government, Sub-directorate of Public Health of Gipuzkoa, 20013 San Sebastian, Spain

**Keywords:** hair Mn, in utero exposure, prenatal, neurodevelopment, prospective

## Abstract

*Background*: Manganese (Mn) is an essential micronutrient for humans, the diet being the main source of exposure. Some epidemiological studies describe a negative association between prenatal Mn and later neuropsychological development, but results are inconsistent. The aim of this study was to explore the association between prenatal Mn exposure and neuropsychological development assessed at 4 years of age. *Methods*: Study subjects were 304 mother-child pairs from the Gipuzkoa cohort of the INMA (Environment and Childhood) Project. Mn was measured in newborns’ hair. Children’s neuropsychological development was assessed at 4 years of age using the McCarthy Scales of Children’s Abilities. Multivariate linear regression models were built. Stratified analysis by sex was performed. Generalized additive models were used to assess the shape of the relation. *Results*: The median Mn concentration in newborns’ hair was 0.42 μg/g (95% CI = 0.38, 0.46). The association between Mn levels and the neuropsychological development was not statistically significant for the general cognitive scale (β [95% CI] = 0.36 [−5.23, 5.95]), motor scale (β [95% CI] = 1.9 [−3.74, 7.55]) or any of the other outcomes. No sex-specific pattern was found. The best shape describing the relationship was linear for all the scales. *Conclusion*: Our results suggest that prenatal Mn concentrations measured in newborns’ hair do not affect cognitive or motor development at 4 years of age in boys or in girls at the observed Mn levels.

## 1. Introduction

Manganese (Mn) is an essential trace element [1] and an essential micronutrient that plays a critical role in normal growth and development, having an impact on body growth, immune function, enzymatic regulation reactions, bone growth and metabolism. It is also needed for proper fetal development [2,3,4]. Nevertheless, overexposure to Mn can also be detrimental to health and accumulation of Mn in the brain may have neurotoxic effects [2,5,6]. 

Mn is naturally found in the environment [7], and for the general population, food is the main source of exposure, since it is present in many foods, especially legumes, cereals, nuts and tea [8]. Air is also a major environmental source of Mn exposure [6,9,10,11,12]. Its presence in the air is most closely linked to the iron and steel industry and the burning of fossil fuels but is also related to its use in the chemical industry. In addition, atmospheric Mn originates from traffic pollution, since it is employed as an additive to improve the octane rating of gasoline [13] and Mn derived from agricultural activities [14,15] should also be considered. The intake of water has been observed to be another source of exposure, especially in regions with high Mn levels in drinking water, such as Bangladesh [16], Japan and Australia [17]. Lastly, tobacco smoke also plays a role as a source of Mn [8], because of the high Mn concentrations in all green leaves [18], and the presumably high absorption of Mn in the lungs. Mothers who smoke have been shown to have increased Mn concentrations in umbilical cord blood [19]. 

Interactions between Fe and Mn are noteworthy when assessing the health effects of Mn intake. While plasma Fe overload significantly decreases the uptake of Mn across the blood-brain barrier, Fe deficiency is associated with increased central nervous system (CNS) burden of this element [20]. The individuals most vulnerable to Fe deficiency are women who are pregnant or of childbearing age, infants and small children [21].

Early research on Mn health effects focused on adult populations highly exposed to Mn mainly due to occupational exposure [8,22]. Reference levels exist for adult population [23] while age-specific exposure limits are not established. More recently, epidemiologic studies have moved toward the examination of potential health effects of lower-level Mn exposure in children through air and drinking water [23]. 

Many cross-sectional studies conducted in infant populations found negative effects of Mn on general cognitive development [10,24,25,26,27]. Negative associations have also been found for specific cognitive functions such as verbal abilities [25,26] or memory [12] and for psychomotor development [28]. Some studies have observed an inverted U-shape relationship between Mn and neurodevelopment [6]. Nevertheless, other studies have failed to find any associations [11,27,29,30].

Prospective studies have had inconsistent findings [23]. Regarding prenatal exposure, Lin et al. [31] describe a negative effect of umbilical cord Mn levels on overall, cognitive and language neurodevelopment at 2 years. Chung et al. [1] found an inverted U-shape association between maternal blood Mn levels at delivery and neurodevelopment at 6 months. Other studies have found a negative effect at 9 months of age but not at later stages [14,32] and Mora et al. [15] found that higher prenatal Mn levels, measured in dentine of deciduous teeth, were associated with poorer behavioral outcomes in school-age boys and girls, but also with better motor function, memory and/or cognitive abilities in school-age boys. Limited and inconsistent evidence of sex-specific neurological effects have been found, generally indicating greater effects in girls both negative [25,27] and positive [33], but also some in boys [34]. 

Despite many studies having analyzed the association between prenatal Mn levels and longer-term neurodevelopment, there is little consistency in the results [23]. Therefore, after having characterized intrauterine Mn exposure, using Mn levels in newborns’ hair as a biomarker of prenatal exposure [19] and considering the high vulnerability of the developing brain, in the present study, we sought to assess whether newborns’ hair Mn levels are associated with neuropsychological development at 4 years of age. Further, we investigated the linearity of the association and possible sex differences. 

## 2. Materials and Methods 

### 2.1. Study Design and Participants

The study participants were mother-children couples of the INMA-Gipuzkoa cohort. The INMA project (from the Spanish terms for environment and childhood, *INfancia y Medio Ambiente*) (http://www.proyectoinma.org) is a prospective multicenter study of mother and child birth cohorts [35]. The main objective of this project is to evaluate the relationship between prenatal and postnatal exposure to environmental pollutants and children’s health from the fetal stage to adolescence [35].

In Gipuzkoa, in the 2006–2008 period, 638 women were recruited in the first trimester of pregnancy. The women were followed during pregnancy and their children were subsequently followed from birth to the fourth year of life. From recruitment to the 4 years old follow up stage there was a 10.65% of participants lost (n = 68) due to: foetal death (n = 4), miscarriage (n = 10), death (n = 2), withdrew (n = 44) and contact lost (n = 8). Of the 612 births, hair samples were collected in 437 newborns (Irizar et al., 2019). Of the 394 children who had a valid neuropsychological evaluation at 4 years of age, 304 also had information on Mn levels in their hair at birth. Therefore, this study included 304 participants for whom we had information on Mn levels in newborn hair and neuropsychological development at 4 years of age (Figure 1). The study was approved by the Ethics Committee of Donostia Hospital (Gipuzkoa) and all mothers signed informed consent forms before inclusion.

### 2.2. Manganese in Hair

A 50 mg sample of newborn hair was taken from the area closest to the occipital region. The samples were washed twice by ultrasonic cleaning and rinsed with ultrapure water. Intra-batch and inter-batch accuracies of reactive targets were estimated at 0.95% and 6.79%, respectively. The accuracy in the concentration range of 4.75–5.68 μg/g (n = 11) was 101.0% (certified reference value of 5.2 μg/g). The analysis was carried out in the Department of Legal Medicine and Toxicology of the University of Granada [36,37,38]. The method has been reported in detail by Irizar et al. [19].

### 2.3. Children’s Neuropsychological Development

A standardized version of the McCarthy Scales of Children’s Abilities (MSCA) was used to assess the children’s cognitive and psychomotor development at 4 years of age [39]. The Basque version of the instrument (MSCA-E) was administered to children whose first language was Basque [40]. The MSCA comprises 18 subtests that yield standardized test scores for six subscales: verbal, perceptual-performance, quantitative, memory and motor scales. The sum of scores on the first three scales provides a General Cognitive Index [40]. In addition, a previously validated Executive Function scale was analyzed [41]. MSCA raw scores were standardized to a mean of 100 and a standard deviation (SD) of 15. All testing was performed by a trained neuropsychologist under appropriate assessment conditions. The neuropsychologist was blinded to any information concerning the child, including hair Mn levels. A total of 4.7% (n = 7) of the test results were excluded from the final analysis due to the poor quality of the child’s assessment (e.g., lack of collaboration or evident tiredness) or to the diagnosis of neurodevelopmental disorders.

### 2.4. Potential Confounding or Predictor Variables

Data on maternal characteristics, including socio-demographic, environmental and lifestyle factors were collected through two questionnaires administered in face to face by interviews in the first and third trimesters of pregnancy.

Information on maternal variables were: education level (up to primary, secondary, university), body mass index (BMI, kg/m^2^) before pregnancy (low weight [<18.5], healthy [18.5–< 25], overweight [25–< 30], obese [≥30]) age at conception (years), parity (0/≥1,), alcohol and smoking consumption during pregnancy (yes/no), and maternal ferritin levels analyzed in the first trimester of pregnancy [(mean (SD) 13.3 (1.5)] by fluoroimmunoassay [42]. Maternal exposure to PM_2.5_ throughout pregnancy and in each trimester was also assessed [43].

Information related to the child’s sex, birthweight (low birth weight < 2500 g yes/no), season of birth, sibling order (first/other) and length of gestation (preterm < 37 weeks yes/no) were retrieved from clinical records. In a subsequent interview when the child was 14 months information on main caregiver, duration of breastfeeding (weeks) nursery attendance, and child second-hand smoke (yes/no) exposure was collected. When the child was 2 years of age was assessed the quality of the family context using the Etxadi-Gangoiti Scale [44]. The subscale of Stimulation of Cognitive and Linguistic Development (SCLD) was also selected because it was strongly associated with cognitive development [45]. The child’s age was also recorded at the time of the neuropsychological evaluation at 4 years of age.

### 2.5. Statistical Analyses

First, a descriptive analysis of the population was performed. Qualitative variables were described in terms of percentages and quantitative variables using mean and standard deviation (SD). It was checked that the assumption of normality was met for the McCarthy scales (Appendix A).

In order to decide a priori which potential confounding or predictor variables needed to be included in our models, we drew a directed acyclic graphs (DAGs) according to current knowledge from the scientific literature (Appendix A). Results identified aminimally sufficient adjustment set that includedmaternal age, smoking during pregnancy, PM_2.5_ and ferritin levels.

In consequence, minimally adjusted regression models were constructed. Then, the model with the best Akaike information criterion (AIC) fit was selected. Thus, final models were adjusted for age at the time of the neuropsychological assessment, sex, sibling order, nursery attendance and maternal education.

Linear regression models were used to analyze the association between hair Mn and all the neuropsychological outcomes. First, hair Mn was included in a continuous manner and later it was analyzed in tertiles (taking lower values as the reference). Subsequently, in order to analyze possible sex differences, analysis was stratified by sex.

Due to the missing data on the Etxadi-Gangoiti Scale scores (15.13%), the models were not adjusted for this variable. On the other hand, sensitivity analyses were performed.

Generalized additive models (GAMs) were fitted to check the linearity of the relationships between Mn and the scales by means of natural cubic splines with one internal knot. In all cases, the linear model was the best for describing the relationship between exposure and outcome.

Statistical analyses were carried out with R, version 3.6.1 (R Core Team, Vienna, Austria).

## 3. Results

An initial analysis was conducted to assess differences in the variables studied between our participants and the whole INMA Gipuzkoa cohort. The only statistically significant differences observed were inmaternal age at conception, participating mothers beingsignificantly older (*p* = 0.029) (Appendix A).

We analyzed data from the 304 children for whom hair samples taken at delivery were available and who had completed a valid neuropsychological assessment (Figure 1). The median Mn concentration in hair was 0.42 μg/g (95% CI = 0.38, 0.46). The geometric mean (GM) was 0.27 μg/g (95% CI = 0.24, 0.31).

Table 1 shows the characteristics of the study population. Regarding maternal characteristics, the women’s mean age at conception was 31.7 years, and 48% of them had university education, 76% had a healthy BMI before pregnancy, 55% were primiparous, 6.91% consumed some alcohol during pregnancy and 10% smoked during pregnancy. Regarding children, 50% were girls, 3.6% had a low-birth weight, and 2.6% were born preterm. They were breastfed for a mean of 30 months. At 12 months of age, 46% attended nursery and 14% were exposed to secondhand smoke. At the 4-year follow up, the children’s mean age was 4.48.

Regression models did not identify any statistically significant associations between newborns’ hair Mn levels and neurodevelopment at 4 years of age (Table 2). Regression models for cognitive outcomes showed that for each 1 µg/g increase in Mn, scores increased on the general cognitive scale by 0.36 [95% CI = −5.23 to 5.95], on the perceptive manipulative scale by 2.24 [95% CI = −3.27 to 7.76], on the executive function scale by 0.1 [95% CI = −5.46 to 5.65], on the general motor scale by 1.9 [95% CI = −3.74 to 7.55], on the gross motor scale by 3.57 [95% CI = −2.1 to 9.23], and decreased on the verbal scale by 0.93 [95% CI = −6.71 to 4.85], on the quantitative scale by 0.43 [95% CI = −5.92 to 5.06], on the memory scale by 0.39 [95% CI = −6.16 to 5.38], andon the fine motor scale by 0.94 [95% CI = −6.23 to 4.36].When analyzing the association in tertiles, regression models again showed no significant associations between hair Mn and the outcomes of interest (Table 2).

Finally, the stratified analysis showed no statistically significant association in either boys or in girls, when analyzing hair Mn levels continuously or in tertiles (Table 3).

In the sensitivity analysis, the results did not vary substantially. Specifically, the inclusion of quality of family context had very little influence, if any, on the regression coefficients for the association between hair Mn and child outcomes, suggesting that this variable was not a significant confounder in our sample (Appendix A).

Results for the GAM analysis showed linearity in all cases (Appendix A). To illustrate this, the models for the General Cognitive and General motor scales are plotted in Figure 2.

## 4. Discussion

In our study, no associations were found between children’s hair Mn levels at birth and neurodevelopment at 4 years of age and. Linear associations were observed between children’s hair Mn levels at birth and all the cognitive and psychomotor scales and no significant patterns of sex-specific associations were found.

Relatively few studies have prospectively examined potential associations between prenatal exposure to Mn and neurodevelopment [1,14,15,31,32,46]. Moreover, different matrices have been used to assess Mn concentrations. Our results are in agreement with those of Takser et al. [32], who found no correlation between Mn measures at birth (from hair and umbilical cord blood samples) and the McCarthy scales general cognitive index at 3 or 6 years. Nonetheless, the concentration of Mn found in hair in their study (Mn GM of 0.75 μg/g [range 0.05–13.33]) was higher than that found in our cohort. To our knowledge, it is the only other study that has measured Mn in newborns’ hair.

Other studies that have analyzed prenatal Mn using dentin as a biomarker of chronic exposure have also found no association with cognitive or psychomotor development during childhood [14,15]. On the other hand, while Erikson et al. [46] found no differences in cognitive ability or psychomotor development, they reported that prenatal Mn levels were positively associated with behavioral outcomes: higher levels of disinhibition (36 months), impulsivity (4.5 years), externalizing and internalizing problems (1st and 3rd grades) and disruptive behaviors (3rd grade).

Regarding studies that have used measures of short-term prenatal exposure, Chung et al. [1] found that associations between maternal blood manganese and neurodevelopment followed an inverted U-shape dose-response curve, but in a recent study conducted with our cohort, no significant association has been found between maternal blood Mn and child cognitive or psychomotor development at 14 months of age [47]. Further, as mentioned in the introduction, other authors have found a negative association between prenatal Mn levels (umbilical cord) and neurodevelopment [31].

Findings are inconsistent across biomarkers regarding associations with mental and psychomotor development [30]. In our study, Mn exposure was measured through hair Mn concentration which, among biomarkers of Mn exposure, is considered a good indicator of chronic environmental exposure [48]. In a recent systematic review, authors concluded that with the evidence available it is not possible to draw definitive conclusions concerning adverse effects of early-life Mn exposure on neuropsychological development, and highlighted the need for more prospective studies to establish a causal link [23].

Our results show linear associations with all the cognitive and psychomotor scales. Similarly, studies using chronic exposure measures (hair or dentine Mn levels) also indicate a linear association between prenatal Mn and long-term cognitive outcomes [15,49]. Gunier et al. [14] described a linear association with cognitive development but a non-linear association with psychomotor development, while other authors proposed an inverted U-shape association between maternal blood Mn and neurodevelopment [1] and some studies have suggested that there may be a threshold value above which there is an association with neurodevelopment [1,31].

In our study, we observed no significant sex-specific patterns of association. Our results suggest a trend in the association between Mn and cognitive scales that is negative for boys and positive for girls. In contrast, the association for the motor scale seems positive for boys but not for girls. Similarly, Gunier et al. [14] did not find an interaction by sex between prenatal Mn levels and mental or psychomotor development at 6, 12 or 24 months. On the other hand, Takser et al. [32] observed a negative association with hand skills in boys and Mora et al. [15] found that higher prenatal Mn levels were associated with better cognitive and motor outcomes in boys only. A possible explanation for the differences found in previous studies between boys and girls are the associated mechanisms of absorption and excretion and interactions with hormones and neurotransmitters [50]. Nevertheless, more epidemiological and animal studies are needed to characterize these potential biological differences between males and females.

The inconsistencies between our results and some previous longitudinal studies could be explained by various factors. Firstly, the use of different matrices to measure Mn exposure. Results are generally discordant between studies that have used blood Mn, hair Mn or tooth dentin Mn levels as a biomarker for Mn exposure [23]. Studies of infants and toddlers most frequently measure the Mn in hair, cord blood, tooth enamel, maternal or child blood or dentin and findings are inconsistent across these biomarkers in relation to associations with mental and psychomotor development [30]. Besides, differences in neurodevelopmental assessment instruments and ages of evaluation may affect the results. The timing of exposure, during pregnancy, may also be critical for Mn neurotoxicity, because the susceptibility of the brain to toxic insults is known to change during different phases of neurodevelopment [5]. Finally, the differences in the Mn concentrations found between studies hinder comparability. Our Mn concentrations are low and could be considered to be within the normal range, in which this compound behaves as an essential element.

This study has some limitations. On the one hand, the sample size, although it is adequate for this type of study, is relatively small and considering the low Mn levels, could be limiting our statistical power. On the other hand, although we were able to assess numerous confounding factors (such as family context, and ferritin or PM2.5 levels), there could be residual confounding in the relationships between Mn exposure and the results of neuropsychological development for which we have been unable to control. In addition, the outcomes studied have been cognitive and psychomotor development and not others possibly related to Mn exposure, such as behavioral outcomes.

Despite its limitations, the present study also has considerable strengths including its longitudinal design, use of comprehensive neurodevelopmental assessments, information on a wide variety of potential confounders, and use of a good indicator of chronic Mn environmental exposure. Considering family contexts have an important influence on children’s psychological development [51] and that effects can be masked by this factor [32], we consider that having measured family context is a key strength of this study. As expected, as no association was found between Mn and neurodevelopment, the inclusion of family context did not change the results. Although it is important to consider all the possible socio-environmental confounders [52], as far as we know, no studies have analyzed the family context among the covariates in their analysis. Further, hair Mn level is considered a good indicator of chronic environmental exposure [48] and therefore a good measure for exploring possible long-term neurodevelopmental effects. Previous studies determining Mn levels have more often beenin older children [30]. Some authors have suggested that Mn levels remain similar throughout childhood [28,32] given that the main Mn sources do not change, it could be reasonable to compare Mn hair levels during childhood with our data. Doing this, Mn concentrations in this study are seen to be similar to those reported in children’s hair in numerous child cohort studies [9,24,27,28,29,37,53] this suggesting that our measure is reliable.

## 5. Conclusions

In conclusion, this study does not show any significant associations between neuropsychological development at 4 years of age and prenatal exposure to Mn. Further, no consistently different sex-specific patterns were detected. Finally, a linear association is observed between exposure and outcome.

It is possible that the prenatal Mn levels observed in our sample are lower than a threshold at which Mn causes neuropsychological adverse effects. These findings therefore provide information to the body of scientific knowledge about which prenatal Mn levels are likely not detrimental and about the window of susceptibility to Mn exposure. Nevertheless, similar epidemiological studies are necessary in order to improve our knowledge about the safe range of this compound in the prenatal period and the possible sex-specific effects.

## Figures and Tables

**Figure 1 ijerph-17-01665-f001:**
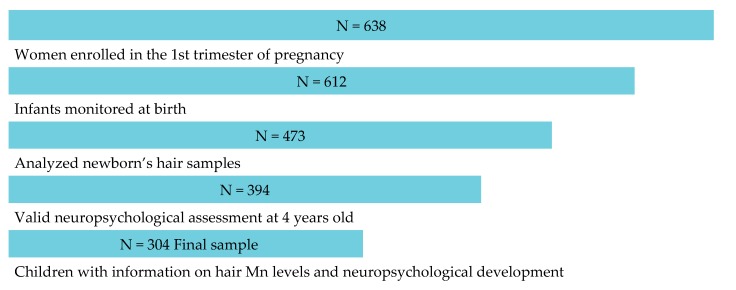
Flow of participants through the study and final sample.

**Figure 2 ijerph-17-01665-f002:**
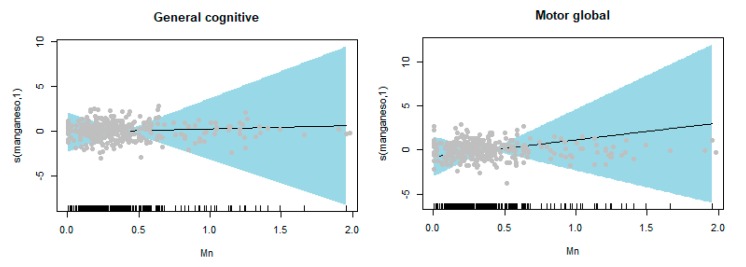
Generalized additive models for the general cognitive and general motor scales.

**Table 1 ijerph-17-01665-t001:** Characteristics of the study population.

		N (%)	Mean (sd)	Number of Missing Data
**Maternal characteristics**				
Age			31.67 (3.3)	0(0%)
Educational level	Primary	38 (12.5%)		1 (0.33%)
	Secondary	120 (39.47%)		
	University	145 (47.7%)		
Body mass index, kg/m^2^	Underweight (<18.5)	13 (4.28%)		0 (0%)
	Normal weight (>18.5–25)	231 (75.99%)		
	Over weight (26–30)	45 (14.8%)		
	Obese (≥30)	15 (4.93%)		
Parity	0	167 (54.93%)		0 (0%)
	≥1	116 (45.07%)		
Alcohol consumedin pregnancy	No	271 (89.10%)		2 (0.66%)
	Yes	21 (6.91%)		
Smoking in pregnancy	No	266 (87.5%)		7 (2.3%)
	Yes	31 (10.20%)		
Ferritin level (μg/L)			34.54 (26.11)	1 (0.33%)
PM_2.5_ in pregnancy (unit)			16.98 (2.43)	22 (7.24%)
**Child characteristics**				
Sex	Male	153 (50.33%)		0 (0%)
	Female	151 (49.67%)		
Low birth weight (<2500 g)	No	292 (96.05%)		1 (0.33%)
	Yes	11 (3.62%)		
Season of birth	Winter	87 (28.62%)		0 (0%)
	Autumn	50 (16.45%)		
	Spring	88 (28.95%)		
	Summer	79 (25.99%)		
Sibling order	First	169 (55.59%)		0 (0%)
	Other	135 (44.41%)		
Preterm (<37 weeks)	No	294 (96.71%)		
	Yes	8 (2.63%)		
Breastfed (weeks)			29.98 (20.20)	12 (3.95%)
Nursery (14 months)	No	150 (49.34%)		13 (4.28%)
	Yes	141 (46.38%)		
Passive smoking (14 months)	No	262 (86.18%)		0(0%)
	Yes	42 (13.82%)		
Etxadi-Gangoiti Scale (at 2 years)			73.71 (9.78)	46 (15.13%)
Age at assessment (at 4 years)			4.48 (0.12)	3 (0.99%)

**Table 2 ijerph-17-01665-t002:** Linear regression models for the association between Mn and the McCarthy scale scores. Change in McCarthy scale scorefor1 μg/g increase in Mn and by tertiles of the Mn distribution.

Beta (CI 95%)								
		For a 1−Point Increase in Mn Levels	*R* ^2^	*AIC*	Tertile 1 [0.0065–0.2316]	Tertile 2 (0.2316–0.4093]	Tertile 3 (0.4093–2.1658]	*p for Trend*
General cognitive	Model 1	−1.05 (−6.72, 4.62)	*0.011*	*2272.542*	1 (Ref.)	0.18 (−4.26, 4.61)	−0.48 (−4.97, 4.01)	*0.834*
	Model 2	0.36 (−5.23, 5.95)	*0.144*	*2128.497*	1 (Ref.)	0.53 (−3.85, 4.9)	0.12 (−4.29, 4.54)	*0.954*
Verbal	Model 1	−1.78 (−7.45, 3.88)	*0.014*	*2271.846*	1 (Ref.)	0.86 (−3.58, 5.29)	0.48 (−4.01, 4.97)	*0.833*
	Model 2	−0.93 (−6.71, 4.85)	*0.100*	*2145.329*	1 (Ref.)	1.58 (−2.94, 6.1)	0.74 (−3.82, 5.3)	*0.744*
Perceptive−manip.	Model 1	0.77 (−4.75, 6.28)	*0.016*	*2257.169*	1 (Ref.)	−0.41 (−4.72, 3.9)	−0.48 (−4.84, 3.89)	*0.829*
	Model 2	2.24 (−3.27, 7.76)	*0.120*	*2121.771*	1 (Ref.)	−0.57 (−4.9, 3.75)	0.23 (−4.13, 4.6)	*0.919*
Quantitative	Model 1	−2.21 (−7.88, 3.47)	*0.009*	*2272.823*	1 (Ref.)	−0.56 (−5, 3.88)	−1.07 (−5.57, 3.42)	*0.638*
	Model 2	−0.43 (−5.92, 5.06)	*0.160*	*2119.094*	1 (Ref.)	−0.11 (−4.41, 4.19)	−0.09 (−4.43, 4.24)	*0.966*
Memory	Model 1	−1.3 (−7.19, 4.58)	*0.006*	*2292.962*	1 (Ref.)	1.26 (−3.35, 5.86)	0.83 (−3.83, 5.49)	*0.726*
	Model 2	−0.39 (−6.16, 5.38)	*0.132*	*2144.533*	1 (Ref.)	1.68 (−2.83, 6.19)	1.23 (−3.32, 5.78)	*0.591*
Executive function	Model 1	−1.43 (−7.05, 4.2)	*0.014*	*2267.655*	1 (Ref.)	−0.23 (−4.63, 4.17)	−0.27 (−4.73, 4.18)	*0.904*
	Model 2	0.1 (−5.46, 5.65)	*0.120*	*2125.193*	1 (Ref.)	0.57 (−3.78, 4.92)	0.61 (−3.78, 4.99)	*0.784*
Motor global	Model 1	0.33 (−5.18, 5.84)	*0.023*	*2256.665*	1 (Ref.)	0.26 (−4.05, 4.57)	−0.16 (−4.52, 4.2)	*0.943*
	Model 2	1.9 (−3.74, 7.55)	*0.060*	*2133.311*	1 (Ref.)	0.58 (−3.84, 5.01)	1.12 (−3.34, 5.57)	*0.621*
Gross motor	Model 1	3.25 (−2.44, 8.93)	*0.038*	*2273.539*	1 (Ref.)	2.21 (−2.23, 6.66)	2.24 (−2.26, 6.74)	*0.326*
	Model 2	3.57 (−2.1, 9.23)	*0.097*	*2135.314*	1 (Ref.)	3.39 (−1.03, 7.81)	3.42 (−1.04, 7.87)	*0.130*
Fine motor	Model 1	−2.99 (−8.36, 2.37)	*0.013*	*2241.805*	1 (Ref.)	−1.98 (−6.17, 2.21)	−2.66 (−6.9, 1.59)	*0.218*
	Model 2	−0.94 (−6.23, 4.36)	*0.135*	*2100.425*	1 (Ref.)	−2.76 (−6.89, 1.38)	−1.98 (−6.14, 2.19)	*0.346*

Model 1: adjusted for maternal age, smoking during pregnancy, PM_2.5_ and Fe levels in pregnancy. Model 2: adjusted also for age at the time of the test, sex, sibling order, nursery at 12 months and maternal educational level.

**Table 3 ijerph-17-01665-t003:** Linear regression models for the association between Mn and McCarthy scale scores. Change in the McCarthy scale scorefor 1 μg/g increase in Mn and bytertiles of the Mn distribution, stratified by sex of the child.

			Beta (CI 95%)			
		Fora 1 Point Increase in Mn Levels	Tertile 1 [0.0065–0.2316]	Tertile 2 (0.2316–0.4093]	Tertile 3 (0.4093–2.1658]	*p for Trend*
General cognitive	Boys	0.01 (−7.63, 7.65)	1 (Ref.)	−0.76 (−7.59, 6.08)	1.17 (−5.43, 7.77)	*0.727*
	Girls	4.56 (−4.01, 13.14)	1 (Ref.)	1.32 (−4.39, 7.03)	1.77 (−4.2, 7.75)	*0.551*
Verbal	Boys	−1.53 (−9.47, 6.42)	1 (Ref.)	1.61 (−5.49, 8.71)	3.01 (−3.85, 9.87)	*0.384*
	Girls	2.68 (−6.35, 11.71)	1 (Ref.)	1.14 (−4.86, 7.14)	0.65 (−5.63, 6.93)	*0.826*
Perceptive−manip.	Boys	3.86 (−3.85, 11.58)	1 (Ref.)	−2.7 (−9.61, 4.21)	0.48 (−6.19, 7.16)	*0.892*
	Girls	2.9 (−5.53, 11.33)	1 (Ref.)	0.86 (−4.74, 6.46)	1.76 (−4.1, 7.62)	*0.552*
Quantitative	Boys	−3.53 (−10.66, 3.59)	1 (Ref.)	−2.15 (−8.54, 4.24)	−2.66 (−8.83, 3.51)	*0.392*
	Girls	5.82 (−3.06, 14.69)	1 (Ref.)	1.8 (−4.08, 7.69)	4.33 (−1.83, 10.49)	*0.166*
Memory	Boys	−1.1 (−8.35, 6.14)	1 (Ref.)	0.96 (−5.52, 7.43)	2.56 (−3.69, 8.82)	*0.417*
	Girls	4.09 (−5.54, 13.72)	1 (Ref.)	1.84 (−4.56, 8.25)	2.37 (−4.32, 9.07)	*0.475*
Executive function	Boys	−2.73 (−10.62, 5.16)	1 (Ref.)	−1.93 (−9.01, 5.15)	−1.11 (−7.95, 5.72)	*0.745*
	Girls	6.12 (−2.08, 14.31)	1 (Ref.)	2.34 (−3.1, 7.79)	4.1 (−1.6, 9.79)	*0.153*
Motor global	Boys	4.06 (−3.61, 11.73)	1 (Ref.)	−0.27 (−7.16, 6.62)	2.03 (−4.62, 8.68)	*0.547*
	Girls	0.05 (−8.56, 8.67)	1 (Ref.)	2.18 (−3.52, 7.88)	2.29 (−3.67, 8.26)	*0.436*
Gross motor	Boys	4.05 (−3.85, 11.96)	1 (Ref.)	3.24 (−3.83, 10.31)	3.98 (−2.85, 10.81)	*0.248*
	Girls	2.3 (−6, 10.59)	1 (Ref.)	5.15 (−0.27, 10.58)	4.53 (−1.14, 10.21)	*0.105*
Fine motor	Boys	1.82 (−5.46, 9.1)	1 (Ref.)	−3.9 (−10.39, 2.59)	−1.19 (−7.46, 5.07)	*0.702*
	Girls	−2.39 (−10.78, 5.99)	1 (Ref.)	−2.22 (−7.78, 3.34)	−1.39 (−7.21, 4.43)	*0.617*

Model 2*: also adjusted for age at the time of the test, sibling order, nursery at 12 months and maternal education.

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
