# Peer review of "Prenatal Manganese Exposure and Long-Term Neuropsychological Development at 4 Years of Age in a Population-Based Birth Cohort"

_ijerph, 2020, doi:10.3390/ijerph17051665_

Round 1

Reviewer 1 Report

This is an interesting study where authors tried to explore the association between prenatal Mn exposure and neuropsychological development assessed at 4 years of age. They concluded that prenatal Mn concentrations measured in newborns’ hair do not affect cognitive or motor development at 4 years of age in boys or in girls at the observed Mn levels.

I have two serious concerns on the study.

There is selection bias as from the initial number of 638, authors ended up to analyze data from 304. Authors should clearly state if they had permission to publish selected data fro the INMA-Gipuzkoa cohort. It is also strange as authors report results from a cohort 12-14 years ago. They should also provide the relevant numbers of permissions. Although methodology and reporting of results are adequate, I would suggest authors to provide with a systematic review and then based on the possible meta-analysis, as they already have stated that there are studies, to find through this, the appropriate sample size and then based on this, to include the proper number of children/women, as it seems that their trial is a retrospective one, which offers through its limitation little on the raised issue.

Reviewer 2 Report

In line 110 you define an accuracy of 101%. It´s imposible. It´s not clear causes of lost in the cohort. You began with 638 women and your final sample is 304, you must to describe and analize posible bias introduced in the stdudy. How do you measure  PM2.5 pregnancy ? What means unit? All the children receive breastfeeding? In order to support your conclussion I suggest to calculate statistical power. Line 224 must be reviewed "Linear associations were observed for all scale (between?)" Clarify the association. 

Reviewer 3 Report

General Comment

This study is aimed on the manganese levels presence in the hair of newborns in a population based-birth cohort and evaluates the exposure within a long-term neuropsychological development.  

Specific Comments

-   The authors mentioned in the introduction the importance of manganese as an essential micronutrient, but it would be appreciable informing what’s the relevance of micronutrients in the normal growth and development. In the line 43-45, the phrase “The intake of water has been observed to be another source of exposure, especially in regions with high levels in drinking water, such as…” is confuse. I supposed that the “high levels” here is referenced to Manganese, right? Still in the introduction, the authors mentioned “low” and “high” levels of manganese, but in terms of concentrations (numbers), what is considered low and high?

-    In the methods, the authors showed the first number of enrolled subjects involving 638 participants, and at the end, the final sample of 304. What was/were the exclude criteria?  

Round 2

Reviewer 1 Report

I think that serious efforts and amendments have been made.

Mild improvements in terms of reporting would improve the quality of the paper.